# Metabolic Engineering of Wine Strains of *Saccharomyces cerevisiae*

**DOI:** 10.3390/genes11090964

**Published:** 2020-08-20

**Authors:** Mikhail A. Eldarov, Andrey V. Mardanov

**Affiliations:** Institute of Bioengineering, Research Center of Biotechnology of the Russian Academy of Sciences, 119071 Moscow, Russia; eldarov@biengi.ac.ru

**Keywords:** winemaking, yeast, strains, alcohol fermentation, metabolic engineering, genomic editing

## Abstract

Modern industrial winemaking is based on the use of starter cultures of specialized wine strains of *Saccharomyces cerevisiae* yeast. Commercial wine strains have a number of advantages over natural isolates, and it is their use that guarantees the stability and reproducibility of industrial winemaking technologies. For the highly competitive wine market with new demands for improved wine quality, it has become increasingly critical to develop new wine strains and winemaking technologies. Novel opportunities for precise wine strain engineering based on detailed knowledge of the molecular nature of a particular trait or phenotype have recently emerged due to the rapid progress in genomic and “postgenomic” studies with wine yeast strains. The review summarizes the current achievements of the metabolic engineering of wine yeast, the results of recent studies and the prospects for the application of genomic editing technologies for improving wine *S. cerevisiae* strains.

## 1. Introduction

For thousands of years, saccharomycetes have been used by humans to produce-wine, bread, beer, and other fermented foods [1,2]. The basis of traditional winemaking is wine fermentation, carried out by the yeast microflora of grapes, wort, and wine-making equipment [3]. Modern industrial winemaking over the past decades is based on the use of starter cultures of specialized wine strains [4]. Commercial wine strains selected as a result of long breeding work have a number of advantages over natural isolates, such as high fermentation efficiency, resistance to high concentrations of alcohol, sugar, sulfite, low temperatures, etc. [5,6]. It is the use of commercial strains that ensures the sustainability and reproducibility of industrial wine-making technologies, guaranteeing the stability of the quality of the resulting drink, which makes it possible to annually receive around 250 million hectoliters of wine worth more than $30 billion [7].

More than two hundred commercial strains of wine yeast available on the world market are actively used by winemakers to produce different types, varieties and brands of wines. Nevertheless, growing competition in the world market, increasing consumer demands for the quality of wine, its useful properties stimulate winemakers and biotechnologists to search for new cultures and technologies [6].

Starting from the 1990s, classical strain improvement methods (CSI) based on the repeated alternation of successive stages of mutagenesis and selection [8] have increasingly been used to obtain starter cultures of wine strains. These methods are quite lengthy and time-consuming, require screening of a significant number of isolates, and they have now been replaced by adaptive or directed evolution methods based on the selection of candidate strains based on the trait embedded in the selection scheme [9,10,11].

The adaptive laboratory evolution (ALE) is a technique of strain optimization that assumes serial or continuous culturing of a particular yeast strain for many generations under selective pressure, such as high ethanol content, high osmolarity etc., thus directing the accumulation of mutants with desired phenotype. As compared to stochastic and laborious CSI techniques, ALE methods are more targeted and convenient [12]. The power of this approach towards optimizing wine yeast is exemplified by generation of strains with altered production of important metabolites (ethanol, glycerol, succinic, and acetic acid) and more rapid sugar utilization [10], strains with increased sulfite tolerance and glycerol accumulation [11], strains with improved resistance towards KCL-induced osmotic stress with increased glycerol and reduced ethanol content [13], as well as enhanced viability and resveratrol production [14].

Finally, the rapid progress in the field of yeast genomics, systems biology, genetic engineering in recent decades have opened up new possibilities for creating new wine strains based on the knowledge of the molecular nature of the target trait or phenotype [15,16,17,18,19]. Unlike the “random” CSI methods, the methods of genetic engineering and directed genomic modification are targeted, i.e., precisely aimed at changing a specific target locus without affecting other sites and without affecting the remaining characteristics of the strain. The possibilities of successfully improving the properties of wine strains by metabolic engineering methods have been demonstrated in dozens of works, however only 2 GM strains of wine yeast are officially registered and approved for commercial use [20,21].

The ML01 strain is capable to perform malolactic fermentation (MLF) simultaneously with alcoholic fermentation due to the presence of integrated copies of a yeast malate permease gene and a bacterial malolactic enzyme gene [20]. During MLF the tart tasting malic acid is converted to softer tasting lactic acid [22] that is essential to provide a smooth round mouthfeel to wines.

The ECMo01 strain contains and additional copy of the *DUR1,2* amidolyase gene [21]. Wine produced with this strain have significantly reduced ethyl carbamate (EC) content, that is important for their nutritional safety. Details of strain construction are provided in Section 2.6.

Certain barriers to the widespread use of GM technologies for creating wine strains are associated with regulatory restrictions and negative public perception in many countries of GM technology [1]. Many of these limitations could be overcome through the correct use of genomic editing techniques.

A number of excellent reviews devoted to various aspects of the metabolic engineering of wine and other industrial yeast strains have been published in recent decades [5,15,17,23,24,25,26]. This review briefly summarizes recent achievements in the development of GM wine yeast strains enabling to improve winemaking technologies, obtain wines with refined nutritional and sensory properties. Also, the results of recent research and the prospects for the application of genomic editing technologies to improve wine and other industrial *S. cerevisiae* strains are discussed.

## 2. Directions of Metabolic Engineering of Wine Yeast Strains

The goals of genetic manipulation of wine strains are aimed at improving such characteristics as the speed and completeness of fermentation, the efficiency of wine processing, preventing the growth of foreign microflora, wine spoilage, refinement of sensory characteristics, nutritional value, including reducing the concentration of harmful and increasing useful compounds, etc. (Table 1) [5,17,24,25,27].

### 2.1. Wine Fermentation and Processing Efficiency

One of the serious problems of winemaking that leads to wine spoilage is “sluggish” or “stuck” fermentation [28]. A common cause of “stuck” fermentation is a lack of yeast assimilable nitrogen (YAN) in the composition of grape juice. Nitrogen is a critical grape nutrient for yeast growth and fermentation activity, affecting various metabolic processes, such as protein synthesis etc. and required to support efficient sugar uptake and catabolism [29]. An imbalance in sugar/nitrogen ratio may lead to a high turnover rate of sugar transporters, resulting in a loss of sugar uptake capacity by the cells in nitrogen-deficient must [30]. YAN deficiency may also lead to accumulation of undesired volatile thiols [31].

Depending on grape variety, ripeness, climate, soil etc. YAN content in grape juice may vary from 50–350 mg/L. In general, it is considered that 120–140 mg N/L is sufficient to complete the fermentation of 200 g/L of sugar [30]. However, kinetics of YAN consumption shows significant strain variation that can be explained by the presence of mutations, affecting the functioning and expression levels of different permeases.

To fill YAN deficiency, various inorganic ammonium salts, such as diammonium phosphate, are used. However, excessive addition of such salts can also lead to a carbon–nitrogen imbalance and adversely affect the quality of wine [30], for instance, supporting the growth of wine spoilage microflora. Therefore, overcoming the nitrogen barrier of wine fermentation by genetic methods and the search for high nitrogen efficient (HNE) strains capable of efficient sugar fermentation at reduced YAN content is an urgent task.

To identify genes whose deletion accelerates fermentation under conditions of YAN limitation a group of authors from the Australian Wine Research Institute analyzed the deletion collection of the haploid wine strain AWRI 1631 under microvinification [32] conditions. Among the 15 mutants detected in this screen, the deletion of the *MFA2* gene of the a-factor yeast sex pheromone had the greatest effect (Table 2).

*MFA2* gene together with paralogous *MFA1* gene encode yeast mating pheromone a-factor–an extracellular signaling peptide of 36–38 amino acids produced by α-haploid yeast cells [58] by a multistep pathway including C-terminal prenylation, N-terminal proteolysis and non-classical export through the ABC transporter *STE6* [59]. Many a-factor intermediates are membrane bound, and in this respect, differ from α-factor intermediates and mature peptides, that are hydrophilic short secreted peptides produced through proteolytic processing of the precursor containing tandem copies of mature 13-aminoacid α factor [60]. In the haploid AWRI1631 (MATα) strain *MFA2* gene is expressed much more efficiently than *MFA1* gene and deletions of these genes resulted in different phenotypes. While *MFA2* gene deletion resulted in a culture with significantly increased (34%) sugar catabolism compared to the parent, deletion of *MFA1* gene did not have such effect. Authors speculate that the positive effect of *MFA2* deletion on sugar consumption under nitrogen limitation may be explained by reduced energy “waste”, but conclude that further work is needed to determine the basis for the link between these genes and fermentation [32].

In another work, the search for HNE strains was carried out using the transposon library of a wine yeast derivative [33]. Deletion of the *ECM33* gene resulted in the shortest fermentation duration (up to 31% reduction) in either a synthetic medium or grape juice (Table 2). Under laboratory conditions, the Δ*ecm33* strain did not differ from the WT strain in nitrogen utilization, cell viability or biomass yield, but was more sensitive to Congo Red and Calcofluor White dyes, known inhibitors of cell wall chitin biosynthesis. Ecm33p is a GPI-anchored cell-wall protein, implicated in efficient glucose uptake, apical bud growth. Transcriptome analysis of the Δ*ecm33* strain suggests Ecm33 is a negative regulator of *SLT2* and *HOG1* genes encoding MAP-kinases involved in cell-wall integrity and high-osmolarity growth signaling pathways. The superior performance of the Δ*ecm33* strain during fermentation may be explained by a more robust cell wall, providing resistance to harsh fermentation conditions as well as by more efficient nitrogen utilization due to the upregulation of several genes of the central nitrogen metabolism [33].

Thus, the need for a further detailed systematic study of the regulation of wine fermentation in yeast under nitrogen starvation conditions is obvious in order to select the optimal solution for this important task.

A useful technological characteristic of wine strains is the ability to prevent the formation of protein turbidity in the production of white wines [61]. Stabilization against protein haze is best achieved using wine strains capable to secrete part of their cell wall glycoproteins. Such strains have other beneficial properties for winemakers due to positive effects of released mannoproteins on aromatic and sensory wine properties, growth of lactic bacteria and MLF etc. [62]. Overexpression of two mannoproteins-Hpf1, Hpf2 (haze protective factors) encoded by the *YOL155c* and *YDR055w* genes in laboratory strain S288C under the control of *GAL1* promoter (fermentation in chemically defined grape juice with 2% galactose) led to more than two-fold decrease in haziness (Table 2) [34].

Yeast’s ability to release mannoproteins depends strongly on strain background as exemplified by the study of the effects of *KNR4* gene deletion [35]. Knr4p is a cell-wall protein involved in cell-wall integrity pathway. Authors have deleted two or three *KNR4* alleles in wine strains EC1118 and T73-4 using different yeast selection markers and measured the haze protective properties and fermentation performance of obtained derivatives (Table 2).

While the fermentation performance of T73-4 derivatives was clearly impaired, and these derivatives did not contribute to the protein stability of the wine, the EC1118 derivative with both alleles of *KNR4* deleted released increased amounts of mannoproteins both in vitro and during wine fermentation assays, and the resulting wines were consistently less susceptible to protein haze. The fermentation performance of this strain was slightly impaired, but only with must with a very high sugar content (Table 2).

GM derivatives of VIN13 strain expressing bacterial glycosidases (pectinase, xylanase, glucanase) that destroy the residual polysaccharides of grape juice had useful properties in terms of facilitating the processing and clarification of wine, stabilizing the color and bouquet of the drink [36].

A promising way to increase the tolerance of wine strains to ethanol, osmotic stresses is to increase the expression of genes induced under the conditions of “fermentation stress” [63]. Two such genes, *HSP26* and *YHR087W*, were introduced into wine strains under the control of their own promoters, the *SPI1* gene promoter, induced at ethanol and osmotic stress conditions, or the potent constitutive glycolytic enzyme gene *PGK1* promoter, as part of plasmids or as a stable chromosome-integrated copies [37]. Stress resistance and fermentation efficiency could be increased in a number of cases, but preservation of regulation of these genes under the control of native promoters turned out to be significant in that regard.

### 2.2. “Biocontrol” Strains

Contamination with extraneous microflora presents a serious threat to winemaking because it can slow down the winemaking process, stop it completely, damage wine [64]. The most serious threat in this regard are lactic acid bacteria, fungi, and yeast of the genus *Brettanomyces/Dekkera* [65,66]. The presence of these microorganisms leads to a decrease in the efficiency of wine fermentation, the accumulation of biogenic amines and unpleasant phenolic compounds [3]. Traditional methods of combating unwanted microflora involve the use of natural or synthetic antiseptics, such as organic acids (citric, benzoic, ascorbic, etc.) and salts (potassium sorbate, sodium benzoate, etc.) [67]. A common way to combat the microbial wine contamination is to treat it with sulfur dioxide, which many wine strains are resistant to [68]. However, sulfite excess is undesirable in wine. Therefore, the search for methods of biological control of extraneous microflora is relevant. For these purposes, specially selected yeast strains producing killer toxins [69] can be used. However, the K1, K2, and K28 killer toxins produced by *S. cerevisiae* are active only against this yeast and cannot be used to control the growth of non-saccharomyces (NS) microflora. Several NS yeast species, such as *Kluyveromyces wickerhamii*, *Pichia anomala*, *Pichia membranifaciens*, and *Candida pyralidae* do produce killer toxins, but their activity against *D. bruxellensis* is low [70].

Known genetic engineering approaches consist in creation of microbicidal GM yeast strains producing bacteriocins [71], glucanases and chitinases [72], and recently, endogeneous *S. cerevisiae* antimicorbial peptides (AMPs) [73]. The discovery of these AMPs –as specific fragments of the glyceraldehyde-3-phosphate glyolytic enzyme may be the most intriguing and promising discovery in this respect. After the initial detection of “saccharomycin”, the proteinaceous toxic compound active against *Hanseniospora guillermondi* cells present in the supernatants of *S. cerevisiae* CCCMI 885 cell fermentations [74], the nature of these AMPs as a mixture of specific peptide fragments, derived from Tdh1p and Tdh2/3p isoforms, was established [73]. Recombinant *S. cerevisiae* strains overexpressing corresponing *TDH1* and *TDH2/3* gene fragments were generated and shown promising to produce this biopreservative, active against both *D. bruxellensis* and lactic acid bacteria at large scale [38].

### 2.3. “Low Alcohol” Yeasts

One of the most important and difficult areas of metabolic engineering of yeast is the production of “low alcohol” strains, i.e., strains capable of releasing less ethanol during wine fermentation while maintaining all other biochemical and organoleptic parameters of the resulting beverage. Wines with reduced strength are finding increasing demand among consumers due to a number of factors. A decrease in ethanol concentration has a positive effect on the nutritional value and organoleptic parameters of wine, and better corresponds to various regulatory standards [75].

There are various technological and microbiological approaches to reducing the alcohol concentration in wine [76]. A group of approaches is aimed at the engineering of wine yeast strains with “redirected” metabolic fluxes of central carbon metabolism from ethanol production to the biosynthesis of other metabolites. A whole series of genetic engineering strategies have been tested to obtain such “low alcohol” yeast, but only few have proven suitable for further practical use [44]. Several approaches aimed at the expression of additional heterologous genes that reduce the level of ethanol biosynthesis precursors, for example, *Aspergillus niger* glucooxidase (GOX) [39], *Lactobacillus casei* lactate dehydrogenase (LDH) [77]. Under the action of these enzymes, part of the glucose was converted to gluconic or lactic acids. Nevertheless, the low GOX efficiency under anaerobic fermentation conditions and the negative effect of high concentrations of lactic acid on the sensory wine properties showed the futility of such approaches.

Another approach consisted of the intensive modification of *S. cerevisiae* glucose transporter genes in order to force the obtained GM strains to switch their metabolism from fermentative to respiratory one regardless of glucose concentration in the culture medium [40]. However, due to a sharp decrease in the sugar uptake capability and the elimination of the Crabtree effect, i.e., the ability to rapidly convert sugars to ethanol and carbon dioxide at both anaerobic and aerobic conditions [78], fermentation in such yeasts became incomplete, often “stuck”, and the ethanol content turned out to be unsuitably low for winemaking.

Alternative strategies for producing “low alcohol” yeast were aimed at manipulating the endogenous genes of alcohol dehydrogenases [41], triosephosphate isomerases [42], and pyruvate decarboxylase [43]. Although some of these approaches turned out to be quite effective in terms of redirecting carbon fluxes to the side of glycerol synthesis, the “fermentative” properties of the obtained strains were unsuitable for winemaking [76].

Another option for reducing the ethanol yield included switching part of the carbon metabolism towards the synthesis of Krebs cycle intermediates. Although both overexpression and deletion of the genes of some enzymes involved in oxidating or reducing TCA branches influenced the content of organic acids, the ethanol yield did not change [44].

In order to relieve glucose repression of genes encoding respiratory enzymes, the authors obtained a strain with deletion of genes for the glucose transporter *HXT2* and the regulator *MIG1*. However, the decrease in the level of ethanol in this strain was very slight [44].

Perhaps the most successful and viable strategy for producing “low alcohol” strains is to redirect part of the carbon flux from ethanol to glycerol at the glycolysis stage. Overexpression of the *GPD1* or *GPD2* genes of glycerol-3-phosphate dehydrogenase isoenzymes increased glycerol concentration by more than five-fold [43]. The resulting wine strains reduced the ethanol concentration in Chardonnay wine from 15.6% to 13.3%, in Cabernet Sauvignon wine-from 15.6 to 12% [79]. The disadvantage of the obtained strains is increased acetaldehyde and acetoin content, negatively affecting wine aroma. The known way to eliminate these drawbacks consists in additional overexpression of the aldehyde dehydrogenase *ALD6* and butanediol dehydrogenase *BDH1,2* genes [44].

Encouraging results were obtained by parallel analysis of collections of wine and laboratory strains overexpressing or deficient in several central metabolism genes [45]. The greatest effect (10% decrease in ethanol content) was exerted by moderate overexpression of the *TPS1* trehalose synthase gene while maintaining complete fermentation and a slight increase in glycerol content.

These studies open up new possibilities for developing approaches for obtaining “low alcohol” strains of yeast, but it is clear that the practical implementation of such strains is still a matter of the distant future. The increased formation of glycerol due to alcohol during fermentation leads to a violation of the redox balance, the formation of wines with high glycerol/ethanol ratio and an unacceptable content of other metabolites that have a negative effect on the sensory wine qualities.

### 2.4. Aroma and Taste of Wine

Wine taste and aroma are main characteristics that determine the differences between a huge number of brands and varieties of wines produced worldwide. The chemical composition of wine depends on many factors, determined by the enormous variability of the conditions of both grape growth and winemaking technologies. These include the grape variety, geographical and technological features of its growing, microbial ecology of grapes and fermentation processes, winemaking methods, and when using starter cultures, the individual characteristics of the strain used.

The contribution of yeast to the formation of the aroma of wine can be due to (a) the production of enzymes that convert “aromatically neutral” grape compounds into aromatically active, (b) biosynthesis of hundreds of aromatically active secondary metabolites, i.e., acids, alcohols, esters, polyolols, aldehydes, ketones, volatile sulfur compounds, and volatile phenols, (c) production of ethanol and other solvents that help to extract aromatic compounds from grape solids, and (d) autolysis of dead yeast cells [80].

#### 2.4.1. Volatile Esters

Esters make the most significant contribution to the formation of a characteristic bouquet of wine during fermentation. The composition of the wine contains up to 160 of these compounds formed during alcoholic fermentation. The most significant effects on the aroma and bouquet of wine have acetate esters and fatty acid ethyl esters. Among the acetate esters, ethyl acetate (“fruity” and “tart” aromas), isoamyl acetate (“banana” flavor) and 2-phenylethyl acetate (“honey”, “pink”, “floral” aromas) can be distinguished. In the formation of ethyl esters of fatty acids, the alcohol group is ethanol, and the acyl group is derived from medium chain activated fatty acids. This group includes ethyl hexanoate (pear flavor) and ethyl octanoate (apple flavor).

To modulate the content of volatile esters responsible for fruit aromas in wine, GM derivatives of strain VIN13 with overexpression of alcohol acetyltransferase genes were obtained. Overexpression of the *ATF1* gene increased the content of ethyl acetate, isoamyl acetate, 2-phenyl acetate, and ethyl caproate during fermentation. Overexpression of the *ATF2* gene had a lesser effect. Overexpression of the *EHT1* gene enhanced the content of esters responsible for apple, apricot, and banana flavor in wine [46].

#### 2.4.2. Monoterpenoids

An interesting approach to increase the content of monoterpenoids in wine is to create GM strains of yeast capable of de novo synthesis of these aromatizing agents. Expression of the sweet basil geraniol synthase gene in the wine strain increased the geraniol content to 750 μg/L, which is more than an order of magnitude higher than the sensory perception threshold of the compound. The total content of other terpenoids increased by more than 200 times in comparison with the control [47].

Important aromatic components of Muscat and Riesling grape varieties are monoterpenes. For the full manifestation of their aromatic properties, these compounds must be released from complexes with various wine wort polysaccharides. Researchers from the Institute of Wines Biotechnology, Stellenbosch University (South Africa), obtained derivatives of the VIN13 wine strain producing secreted *Aspergillus awamori* arabinofuranosidase along with *Aspergillus kawachii* β-glucosidase. Wine obtained using a strain expressing both enzymes contained higher concentrations of monoterpenes (citronellol, linalol, nerol, and α-terpenylol) than wine obtained by processing with a commercial enzyme preparation and possessed improved sensory characteristics [48].

#### 2.4.3. Diacetyl Removal

High concentrations of diacetyl give an unpleasant, oily, aftertaste to the wine. Diacetyl is a side prolapse of the valine metabolism and can be converted to acetoin and butanediol by the action of the Bdh1p and Bdh2p Butandiol Dehydrogenases. Due to the coexpression of *BDH1,2* genes in the *S. uvarum* strain, the authors were able to almost halve the concentration of this unpleasant impurity component [49].

#### 2.4.4. “Raspberry” Yeast

A striking example of the successful use of metabolic engineering and synthetic biology methods for the directional change in the aroma-forming characteristics of wine strains is the work devoted to the engineering of the biosynthesis of 4-[4-hydroxyphenyl] butanedione, or frambion [50]. Frambion is a raspberry ketone, the main aromatic phenlipropanoid of some fruits, vegetables, berries, including raspberries, blueberries, grapes. The concentration of frambion in natural raw materials is quite low and the basis for the method of commercial production of this flavor is chemical synthesis. The path of biosynthesis of frambion includes four main stages. The first stage is the production of p-coumaric acid by the conversion of phenylalanine through cinnamate or by direct conversion of tyrosine. The conversion of coumaric acid to raspberry ketone requires three additional steps, including the condensation step between coumaril-CoA and malonyl-CoA. To engineer the biosynthesis of frambion in a wine strain, four synthetic genes encoding enzymes of frambion biosynthetic pathway from thale cress, parsley, rhubarb under the control of the *FBA1* gene promoter were integrated into the H0 locus of the AWRI strain (Figure 1A). *FBA1* encodes fructose 1,6-bisphosphate aldolase, a critical cytoplasmic enzymes required for glycolysis and gluconeogenesis [81] and is induced during growth on non-sugar carbon sources and at late stages of wine fermentation [82].

The resulting strain was able to produce frambion at concentrations of 0.68 mg/L-2 orders of magnitude higher than the threshold level of its sensory detection (0.001–0.01 mg/L) in chardonnay grape juice while retaining the ability to completely ferment wine wort [50].

#### 2.4.5. Resveratrol-Producing Yeast

Resveratrol (RV) is a potent plant antioxidant with multiple beneficial effects on human health and is therefore used in medical, food, and cosmetic areas [83]. In grapevines this stilbene compound is produced as stress metabolite, present predominantly in skins of grape berries [84]. Thus, RV content in red wines is much higher than in white wines [85]. Since RV-enriched wines are of significant nutritional value [86], efforts had been made to develop wine yeast strains capable to produce RV during fermentation of both red and white wines. In higher plants, RV is derived from phenylpropanoid pathway, starting from phenylalnine or tyrosine as RV biosynthetic precursors [87] (Figure 1B).

Recombinant RV production was first shown in laboratory *S. cerevsiae* strain, engineered to express 4-coumaroyl-CoA ligase gene from poplar and resveratrol synthase gene from grapevine [88]. The obtained transfomant was able to produce RV at low levels (around 1 ng/mL) in the form glycoside piceid and only upon feeding expensive precursor- p-coumaric acids. The major breakthroughs in development of yeast strains capable of de novo RV production starting from glucose or ethanol are due to the efforts of the research team from the Novo Nordisk company [51,89]. In the first study authors had reconstructed the TAL pathway of RV biosynthesis in industrial *S. cerevisiae* strain through overexpression of bacterial *TAL* genes, *A. thaliana 4CL* gene and *VST* gene from *Vitis vinifera* [89]. The initial RV production level of about 3 mg/L was raised to around 500 mg/L after the application of complex metabolic engineering and a fermentation optimization strategy aimed at increasing gene expression levels, fluxes of RV biosynthetic precursors, and biomass yield [89]. In another study the PAL pathway for RV production in *S. cerevisiae* was engineered by introducing *A. thaliana AtPAL2*, *AtC4H At4CL2* genes and *V. vinifera VST1* gene in the same CEN.PK102-5B strain under the control of strong constitutive pTEF1 and pPGK1 promoters [51]. Cultures supplemented with phenylalanine were able to produce RV at about 30 mg/lL. This level was raised to a record 800 mg/L in fed-batch fermentations after application of a “pull-push-block” strain engineering strategy that included overexpression of the RV biosynthetic genes, optimization of the electron transfer to the cytochrome P450 monooxygenase, increase in precursor supply, decrease of the pathway intermediates degradation. Moreover, through the introduction of heterologous methyltransferases in the RV platform strain, it was possible for the first time to demonstrate de novo biosynthesis of RV derivatives pinostilbene and pterostilbene, which have better stability and uptake in the human body [51].

### 2.5. Flor Yeast Strains

A special group of winemaking microflora is represented by flor *S. cerevisiae* strains used in a number of traditional technologies for production of biologically aged wines [90,91], such as various varieties of Sherry (Spain), Vin jaune (France), Vernachcha di Oristano (Sardinia, Italy), Samorodnyi dry Tokaj (Hungary) [92].

In the course of prolonged biological exposure under the flor yeast vellum, wine acquires specific taste and aroma characteristics caused by changes in the yeast metabolism from enzymatic to oxidative one [92,93,94]. Genetic, biochemical, and physiological properties of flor yeast associated with their adaptation to the specific conditions of sherry winemaking have been studied in sufficient detail [95]. Using microsatellite analysis, a high degree of phylogenetic relationship of flor strains was established [96]. Comparative genomic analysis revealed numerous genetic differences specific to flor yeast in different pathways of metabolism and cell signaling, such as oxidative metabolism, cell wall biogenesis, stress tolerance, lipid biosynthesis, and ion transport of potentially adaptive value [97,98] The events of gene loss and acquisition specific for flor strains, specific genomic loci distinguishing flor and wine strains, probably positively selected were identified [98,99].

Key flor yeast strain characteristics, such as the ability for efficient vellum formation, resistance to high alcohol and acetaldehyde concentrations, to low pH etc are largely associated with the properties of cell surface proteins.

The key role *FLO11* adhesin gene for flor yeast biofilm formation was proven in experiments on its genetic inactivation [52] or overexpression [100]. Overexpression of genes for several other cell wall proteins, such as *Ccw14p* and *Ygp1p* [53], as well as deletion of the *BNT2* gene encoding one of the vesicular transport proteins [101], can also increase the ability to film formation of flor strains.

Promising targets for further genetic and genetic engineering manipulations with flor strains aiming at improving their biofilm formation ability and resistance to various types of stress are *HSP12* [102] and *HSP150* [103], heat shock protein genes, and *SOD1* and *SOD2* superoxide dismutase genes, genes for enzymes of the gluthathione biosynthesis pathway [54] (Figure 2).

### 2.6. Commercial GM Wine Yeast Strains

Despite numerous and successful attempts to improve the characteristics of yeast wine strains by genetic engineering methods, only two GM strains are officially registered for use in the USA, Canada, and Moldova, which is associated with both conservative winemakers and well-known public prejudices and legislative restrictions on the use of GM technologies for food [104].

Strain ML01 [20] is capable to carry out malolactic fermentation (MLF) simultaneously with wine fermentation, i.e., to turn malic acid into lactic with the release of carbon dioxide and water. The content of malic acid in wine wort can be up to 10 g/l, giving a sharp, tart taste characteristic of young wine. Replacing malic acid with lactic acid leads to a decrease in total acidity, improves taste, aroma, body of wine. MLF is a necessary stage in the technology of red wines and is normally carried out by lactic acid bacteria of the wort, for example, *Oenococcus oeni*. However, these bacteria are very whimsical, i.e., sensitive to inhibitory conditions of wine fermentation (low pH, high ethanol content, lack of nutrients), which can slow down or stop this important technological process. Therefore, a strain of wine yeast capable of simultaneously performing NMB would be of great interest both to winemakers and to consumers.

Strain ML01 obtained on the basis of strain S92 contains two chromosomally integrated genes- the *Schizosaccharomyces pombe mae1* gene encoding malate permease and the *O. oeni* malolactic enzyme gene *mleA*. Both genes are placed under the control of the strong constitutive promoter of the *S. cerevisiae* PGK gene. The strain is able to completely ferment malic acid at 5g/L concentration in the wort within 5 days, without negative effect on the sensory wine properties. Further detailed phenotypic, transcriptome, proteomic analysis showed that strain ML01 is equivalent to the original parental wine strain [20].

Another GM wine yeast strain approved for use in USA and Canada was obtained in order to reduce ethyl carbamate (EC) content in wines [21]. EC is a carbamic acid ethyl ester formed during wine storage. The precursor of EC is urea, an intermediate product of yeast arginine catabolism. EC content in wines may be quite significant (0.01–0.025 mg/L) and increases sharply at elevated temperature. EC is a compound with a possible carcinogenic effect and the EC content in food products is regulated by EU standards.

Strain ECMo01 contains an additional copy of the amidolyase gene *DUR1,2* under the control of the regulatory sequences of the PGK1 gene [21]. Dur1,2p is responsible for the conversion of urea to ammonia and carbon dioxide. In the ECMoO1 strain, the expression of the *DUR1,2* gene is 17 times increased, which leads to a decrease in the urea concentration, and the ammonia produced is utilized as a nitrogen source. The concentration of EC in wine obtained using the ECMoO1 strain was reduced by 90%, while the phenotypic characteristics of the strain are equivalent to the original strain 522.

## 3. CRISPR-Cas for Wine Yeast

For laboratory strains of *S. cerevisiae*, an extensive and diverse set of tools for genetic engineering and directed modification of the genome has been developed quite a long time ago and are widely used for research in the fields of functional genomics, synthetic biology, biotechnology, and metabolic engineering [105]. At the same time, the application of such approaches for industrial strains faces a number of difficulties. These strains are usually polyploids and aneupoloids, poorly sporulate, there are no convenient auxotrophic markers for them, etc. [106].

The use of CRISPR-Cas genome editing systems can successfully overcome these limitations. The first work on the application of the CRISPR-Cas system for *S. cerevisiae* was published back in 2013 [107] and the advantages of this approach for yeast, in which the system of homologous recombination was already well developed, were at first not obvious. However, after overcoming a number of technical difficulties aimed at optimizing the expression and delivery of CRISPR-Cas system components, the system quickly gained popularity and is now successfully used in areas such as multiplex genome engineering, reprogramming transcription, creating synthetic genomes, etc. [108].

Examples of the successful application of CRISPR-Cas systems for industrial yeast strains relate to such aspects as the production of bioethanol from lignocellulosic raw materials, metabolic engineering for the production of vitamins and antibiotics, the improvement of aromatic and taste properties of beer, and a number of others [109,110,111]. From the point of view of food safety, it is fundamentally important that the use of CRISPR-Cas genome editing methods does not carry the risk of introducing foreign genes and genetic elements, markers of antibiotic resistance into the genomes of food yeast strains, i.e., the resulting strains are safe according to regulatory restrictions adopted in some countries.

One recent study describes the use of the CRISPR-Cas system for producing wine strains with reduced urea production. A group of scientists from Canada and Italy constructed derivatives of wine strains EC1118 and AWRI1796 defective in both alleles of the *CAN1* gene [90]. The *CAN1* gene encodes arginine permease, which along with *GAP1* amino acids permease is responsible for the transport of arginine to yeast cells from the culture medium. During the subsequent stages of catabolism, arginine is cleaved by Car1p arginase to ornithine and urea, which is either excreted by Dur4p permease or converted to carbon dioxide and ammonia by Dur1p/Dur2p urea amidolyase. The resulting recombinant strains were characterized by reduced urea production (18–36% compared to the initial ones) under experimental micro-winemaking with the ability to ferment a synthetic substrate, although at a slightly reduced growth rate. The authors believe that further verification of the strains is necessary under the conditions of industrial winemaking. The advantage of introducing a mutation into the *CAN1* gene compared to other methods of modifying arginine utilization pathways is that this technique is less sensitive to fluctuations in the content of nitrogen sources in the wort and less affects the growth parameters of yeast strains [111].

A promising area of application of genome editing methods is the directed change in the pathways of biosynthesis of aromatic compounds. Thus, in a recent work, yeast strains with increased production of phenylethyl acetate (PEA) were obtained using the CRISPR-Cas system [55]. PEA is an important aromatic compound that provides alcoholic drinks a pink and honey flavor. Genetic mapping methods first identified unique alleles of the *FAS2* genes (encodes the α subunit of fatty acid synthase) and *TOR1* (a growth regulator in response to the availability of a nitrogen source), linked to the trait of increased PEA production. Then, using CRISPR-Cas in commercial wine strains, wild alleles were replaced with mutant ones. As a result, the production of PEA increased by 70% [56].

In another work, the CRISPR-Cas system was used to reduce the production of 4-vinyl guaiacol (4VG) in a hybrid *S. pastorianus/bayanus* beer yeast strain [57]. It is known that 4VG is a sharp-tasting phenolic compound that spoils the organoleptic characteristics of beer. Formed from ferulic acid, 4VG is present in beer wort under the influence of yeast decarboxylase Fdc1p. Ale beer yeast strains do not produce 4VG due to the nonsense mutation in the *FDC1* gene. Using the CRISPR-Cas system, the authors introduced a mutation characteristic of ale strains into all four copies of the *FDC1* gene in the lager strain. The result was a strain containing a cis-gene mutation that lacks the ability to produce 4VG and has significant potential for use in the beer industry.

The CRISPR-Cas system is an extremely convenient tool for research in the field of functional genomics of wine strains. Until recently, the vast majority of experiments in the field of functional genomics of yeast were performed using laboratory strains. Nevertheless, according to the latest information from the SGD database (27 June 2020), when classified in terms of gene ontology, a significant number of yeast genes remain “unknown” (in the category “Biological Process”-1768 genes, 2548 genes in the category “Molecular Function” and 1298 genes in the cell compartment category). Such uncertainty is partly determined by the lack of specific conditions in which these genes are important. At the same time, these unknown genes experience regular changes in expression during many technological processes, including at different stages of wine fermentation (see, for example, [112]).

Characteristic changes in the expression pattern of a number of “unknown” genes were revealed in our recent work during the transcriptome analysis of the sherry strain at different stages of film formation [113]. CRISPR-Cas mediated genetic inactivation of “unknown” genes, allele replacement in wine strains of yeast can significantly clarify their role in various winemaking processes, and will help to create strains with improved characteristics. 

## 4. Conclusions

The extensive arsenal of genetic manipulation methods developed for laboratory strains of *S. cerevisiae* can be successfully used in oenology for the metabolic engineering of wine strains. GM strains of wine yeast with targeted changes in various stages of the central or secondary metabolism have proven effective both in optimizing the winemaking process itself and in improving the quality of the resulting wines. Selected examples discussed in the review are shown in Table 2. Despite numerous examples of the successful use of GM strains in experimental winemaking, well-known public prejudices and legislative restrictions hinder the widespread adoption of GM technologies. Promising alternatives to GM technologies are experimental directed evolutionary methods, interspecific hybridization, and selection methods that are already in demand in the wine industry. The rapid progress of research in the field of comparative genomics and systemic biology of wine strains provides unique opportunities for using high-precision genome editing methods to improve the characteristics of wine strains that are free from legislative restrictions. Indeed, the CRISPR-Cas system could soon become the gold standard for new microorganisms suitable for the food industry. However, the ruling of the European Court of Justice adopted in 2018, which essentially equates plants, animals, and microorganisms obtained by genomic editing with genetically modified organisms [114] postpones this prospect until better times.

## Figures and Tables

**Figure 1 genes-11-00964-f001:**
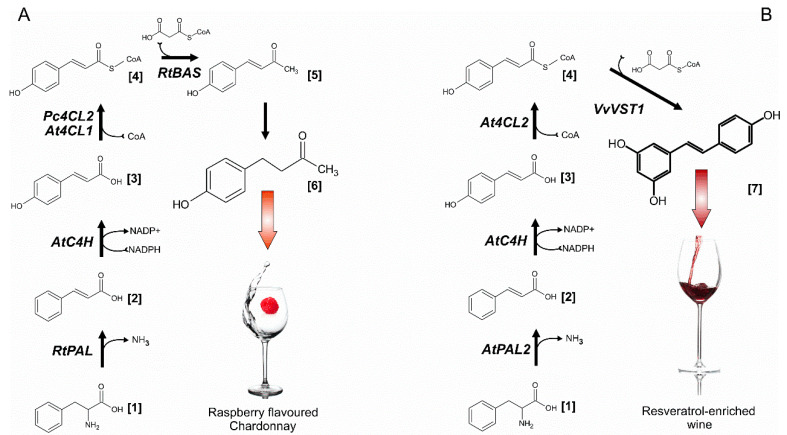
Reconstruction of biosynthetic pathways for production of frambion (**A**) or resveratrol (**B**) in yeast. Metabolite designation: [1]—Phenylalanine, [2]—Cinnamic acid, [3]—p-Coumaric acid, [4]—p-Coumaryl-CoA, [5]—benzalcetone, [6]—frambion, [7]—resveratrol. Enzyme designation: AtPAL2—phenylalanine ammonia lyase from *A. thaliana*, RtPAL- phenylalanine ammonia lyase from *Rhodosporodium toruloides*, AtC4H—cinnamate-4-hydroxylase from *A. thaliana*, At4CL2—p-coumaryl-CoA ligase 2 from *A. thaliana*, Pc4CL2—p-coumaryl-CoA ligase from *Petroselinum crispum*, RpBAS—benzylacetone synthase from *Rheum palmatum*, VvVST1, resveratrol synthase from *V. vinifera.*

**Figure 2 genes-11-00964-f002:**
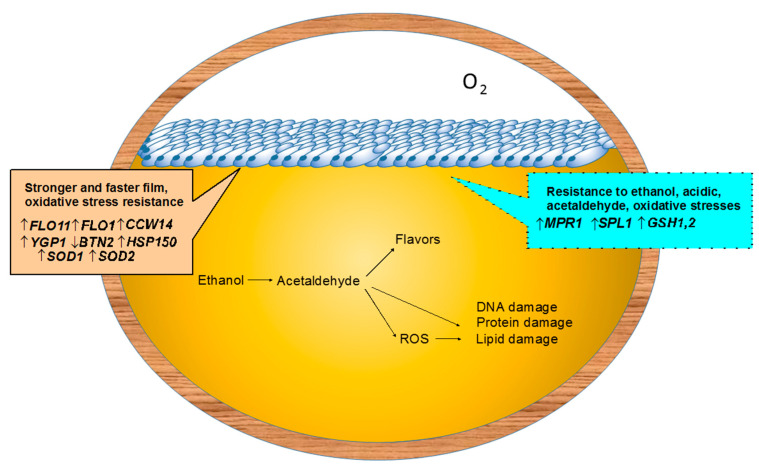
Targets for genetic improvement of flor yeast strains. Explored genes are outlined in beige, promising targets–in blue.

**Table 1 genes-11-00964-t001:** Oenological characteristics of wine strains of *S. cerevisiae*-targets for metabolic engineering.

**Alcohol Fermentation**	
The efficiency of sugar assimilation and the fermentation process itself	Resistance to osmotic and ethanol stresses
Efficiency of nitrogen assimilation	Reduced foam formation
General “endurance” and stress resistance	Moderate biomass accumulation
**The Nutritional Qualities of Wines**	
Increased Resveratrol content	Reduced content of biogenic amines
Reduced Ethyl Carbamate	Reduced alcohol content for low alcohol wines
**Pest Protection**	
Optimum sulfur dioxide production	Optimal antimicrobial enzyme production
Resistance to antimicrobial agents	Optimal antimicrobial peptide production
**Wine Processing Technologies**	
Simplification of wine clarification	Film formation (for technologies of sherry wines)
Compact sediment (for champagne technology)	
**Organoleptic Properties**	
Ability to release aromatic terpenoids	Ability to release and convert aromatic thiols
Increased Glycerol Productiona	Optimized fusel oil production
Reduced Volatile Acid Production	Reduced hydrogen sulfide production

**Table 2 genes-11-00964-t002:** Selected metabolically-engineered yeast strains and their oenology-related phenotypes.

Strain	Genetic Modification	Oenology-Related Trait	Ref.
ML01	Overexpression of *S. pombe mae1* gene *O. oeni mleA* gene	Malolactic fermentation	[20]
ECMo01	Overexpression of *S. cerevisiae DUR1,2* gene	Reduced ethyl carbamate content	[21]
AWRI 1631	Deletion of *MFA2* gene	Improved fermentation efficiency under nitrogen limitation	[32]
C911D	Deletion of *ECM33* gene	Improved fermentation efficiency under nitrogen limitation	[33]
S288C	Overexpression of *S. cerevisiae YOL155c* and *YDR055w* genes	reduced haziness during fermentation	[34]
EC1118	Deletion of *KNR4* gene	reduced haziness during fermentation, retaining good fermentation performance	[35]
VIN13	Overexpression of *Butyrivibrio fibrisolvens end1* gene, *Aspergillus niger xynC* gene	decrease in wine turbidity, increase in colour intensity, increase in phenolic compounds	[36]
VIN13	Overexpression of *Erwinia chrysanthemi pelE* gene, *Erwinia carotovora peh1* gene	decrease in phenolic compounds	[36]
ICV16, ICV27	Overexpression of *S. cerevisiae HSP26* and *YHR087W* genes	Improved Stress resistance and fermentation efficiency	[37]
PYCC 5484	Overexpression of 925–963 segments of *TDH1* and *TDH2/3* ORFs	Secretion of AMPs, inhibiting *D. bruxellensis* growth	[38]
Sigma1278	Overexression of *A. niger GOX* gene	Reduction of sugar content in juice	[39]
V5.TM6 *P.	Overexpression of chimeric *HXT1-HXT7* gene in a *hxt* null strain	decreased ethanol production, increased biomass under high glucose conditions	[40]
MC42	Deletion of *ADH1*, *ADH3*, *ADH4* genes, *ADSH2* gene mutations	66% reduction of ethanol yield, increased glycerol production	[41]
CEN.PK 113-7D	Deletion of *TPI1* gene	Unable to grow on glucose, growth on mixed substrates	[42]
YSH l.l.-6B	Deletion of *PDC2* gene, overexpression of *GPD1* gene	Reduction of glucose catabolism, 6-7-fold increase in glycerol formation	[43]
AWRI1631	*GPD1* overexpression, *ALD6* deletion *	Decreased ethanol production	[44]
BY4742, VIN13	Screening of EOROSCARF deletion collection, weak *TPS* overexpression	10% reduction in ethanol yield, increased glycerol, trehalose production	[45]
CMBS33, BY4742	Analysis of ATF1,2 knockouts in the lab strain, constitutive *ATF1,2* overexpression in lager strains	Reduction in acetate esters production in ATF1,2 deletion strains, enhanced production of volatile esters in overexpression strains	[46]
T73-4	Overexpression of *Ocimum basilicum* (sweet basil) geraniol synthase (*GES*) gene	Increased geraniol production during fermentation, 230-fold increased total monoterpene content	[47]
VIN13	Overexpression of *A. awamori* arabinofuranosidase, *A. kawachii* β-glucosidase.	increased release of citronellol, linalool, nerol and α-terpineol.	[48]
WY1	Overexpression of *BDH1,2* genes	Decreased diacetyl, increased acetoin, butanediol contents	[49]
AWRI	Overexpression of *RtPAL*, *AtC4H*, *At4CL*, *RtBAS* genes for frambion biosynthesis	Frambion production at 0.68 mg/L simultaneously with chardonnay wine fermentation	[50]
CEN.PK 113-7D	Overexpression of *AtPAL2*, *AtC4H*, *At4CL*, *VvVST1* gene for resveratrol biosynthesis, complex strain and cultivation optimization strategy	Yeast-based de novo resveratrol production from glucose at 800 mg/l level	[51]
133d	Overexpression of *FLO11* gene using different promoter variamts	Improved velum formation	[52]
P3-D5	Deletion of *CCW14, YGP1* genes in a flor strain	Impaired velum formation	[53]
FJF206, FJF414, B16	Overexpression of *SOD1*, *SOD2*, *HSP12* in flor strains	increased superoxide dismutase, catalase, gluthathione peroxidase activities, increased oxidative stress resistance, quicker velum formation, slight decrease in ethanol and increase in acetaldehyde content	[54]
EC1118, AWRI1796	Crispr-cas9 mediated inactivation of *CAN1* gene	Reduced ethyl-carbamate formation	[55]
BTC.1D	Crispr-cas9 mediated allele exchange for *FAS2* and *TOR1* genes in wine strain	Increased phenyl-ethyl acetate formation	[56]
W34/70	Crispr-cas9 mediated allele exchange for FDC1 gene in lager strain	Decreased 4-vinyl guaiacol formation	[57]

* Other modifications had non-significant effects.

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
