# Peer review of "Metabolic Engineering of Wine Strains of Saccharomyces cerevisiae"

_genes, 2020, doi:10.3390/genes11090964_

Round 1
Reviewer 1 Report
This paper is well written. I think it only needs a few minor adjustment.
It would be useful if the authors could include a table displaying the yeast stains discussed in the paper. Their key characteristics can be included.
Grammar check should be performed. For example, on line 55, "summarize" should be "summarizes".
Author Response
1) It would be useful if the authors could include a table displaying the yeast stains discussed in the paper. Their key characteristics can be included.
RE: additional table describing yeast strains discussed in the manuscript added
2) Grammar check should be performed. For example, on line 55, "summarize" should be "summarizes".
RE: Article proof-reading had been performed.

Reviewer 2 Report
Mikhail A. Eldarov and Andrey V. Mardanov in the paper entitled "Metabolic Engineering of Wine Strains of Saccharomyces cerevisiae" provide a synthetic description of the most important aspects of the metabolic engineering of wine yeast strains. The authors described both the current achievements of the metabolic engineering of wine yeast and prospects for the application of genome editing technologies for improving S. cerevisiae wine strains and therefore for improving the nutritional and sensory properties of wine. These two aspects are well described based on wide literature data.
Although there are quite a number reports in the scientific literature on various issues of metabolic engineering of wine strains this manuscript presents a slightly different approach and provides a valuable and recent data in this field.
Minor points:
1) The scheme 1 is not of good quality and its quality should be improved for better readability
2) The figure 2: the description should be corrected, it should be clearly identified what is "known" and what is "promising" targets for genetic improvement of yeast strains
3) It would be important to provide an additional information on the metabolic engineering of wine strains of S. cerevisiae towards the synthesis of the resveratrol (it was mentioned in the table 1)
Author Response
1) The scheme 1 is not of good quality and its quality should be improved for better readability
RE: Figure quality was improved
2) The figure 2: the description should be corrected, it should be clearly identified what is "known" and what is "promising" targets for genetic improvement of yeast strains
RE: Figure 2 had been modified according to reviewer’s recommendations.
3) It would be important to provide an additional information on the metabolic engineering of wine strains of S. cerevisiae towards the synthesis of the resveratrol (it was mentioned in the table 1)
RE: we have added a special section describing progress in S.cerevisiae towards resveratrol production.

Reviewer 3 Report
The review article by Eldarov et al. summarizes the current achievements of metabolic and genetic engineering of wine yeast. Though the article is an interesting read, it will benefit the readers greatly if the authors can elaborate certain sections of the manuscript rather than just mention a sentence summarizing the previous findings. Another important thing is that through out the text the gene names are written only in Caps and not italicized. All gene names should be italicized as that is the norm in S cerevisiae. Here are my comments:
Abstract: Lines 12-14: "At the same time,..... new strains and technologies." -- the sentence is very difficult to read and needs to be written in a simplified way.
Introduction: Lines 38-40. How is adaptive or directed evolution methods of selection is advantageous than CSI? In other words why CSI was replaced by these methods. Please explain in details.
Lines 21,27, 71: Please be consistent in spelling "wine-making". It is spelled as winemaking in lines 21 and 71.
Line 47: "Only 2 GM strains of wine...." please provide details of these two strains in the introduction.
Table 1: The authors listed the oenological characteristics. However, for better clarity the authors should provide in table 1 the list of S. cerevisiae genes that can be used as targets for genetic modifications to regulate these listed characteristics.
Line 79. Clarify the role of MFA2 gene. What does this gene encodes for? What makes MFA2 gene of a-factor different from the alpha mating factor?
Lines 80-82: "In another work, the search for genes.... by transposon mutagenesis." -- How lack of nitrogen benefits fermentation? Please discuss.
Line 83: "Deletion of ECM33 gene..." What is the function of ECM33 in yeast. Why is the fermentation time reduced when it is deleted?.. Please clarify
Line 84:.... cell wall damage and osmosis resistance". How cell wall damage and osmosis resistance affects wine-making? Please elaborate.
Line 92: "Deletion of KNR4 gene..." Explain the role of KNR4 in yeast. "to a decrease in protein turbidity under laboratory conditions." Elaborate the conditions.
Please provide a table stating all the yeast strains and their important genotypes and phenotypes for wine-making.
Line 101: were introduced into wine strains under control of various promoter.." Please elaborate what promoters were used in previous studies.
Line 107: "poses" spelling is wrong should be "possess"
Line 121: "for peptides derived from..." please elaborate on the different peptides that in the past derived from endogenous yeast.
Line 145: What is crabtree effect? Please discuss
Lines 158: "glucose transporter HXT2 and regulator MIG1..." proteins need to be written as Hxt2 and Mig1
Line 235: "Rasberry" spelling is wrong should be "Raspberry". The spelling also needs to be corrected in Figure 1. Figure 1 texts needs to be bigger as it is difficult to read.
Line 247: ...promoter of FBA1 gene" -- elaborate the role of FBA1 in yeast.
Line 250: ....the threshold level of its sensory detection"... Please specify the sensory threshold value in the text
Line 260: References [59.60] should be [59,60].
Author Response
1) Another important thing is that through out the text the gene names are written only in Caps and not italicized. All gene names should be italicized as that is the norm in S cerevisiae.
RE: corrected
2) Abstract: Lines 12-14: "At the same time,..... new strains and technologies." -- the sentence is very difficult to read and needs to be written in a simplified way.
RE: abstract had been re-written
3) Introduction: Lines 38-40. How is adaptive or directed evolution methods of selection is advantageous than CSI? In other words why CSI was replaced by these methods. Please explain in details.
RE: explanation of the advantage of adaptive evolution methods is provided
4) Lines 21,27, 71: Please be consistent in spelling "wine-making". It is spelled as winemaking in lines 21 and 71.
RE: corrected
5) Line 47: "Only 2 GM strains of wine...." please provide details of these two strains in the introduction.
RE: We have inserted a paragraph with brief insertion of these strains
6) Table 1: The authors listed the oenological characteristics. However, for better clarity the authors should provide in table 1 the list of S. cerevisiae genes that can be used as targets for genetic modifications to regulate these listed characteristics.
RE: According to your advice and also the advice of reviewer 1 we have inserted additonal table 2 describing yeast strains discussed in the manuscript
7) Line 79. Clarify the role of MFA2 gene. What does this gene encodes for? What makes MFA2 gene of a-factor different from the alpha mating factor?
RE: We have inserted a paragraph with brief description of MFA2, MFA1 genes and proposed mechanism of the influence of MFA2 deletion of fermentation efficiency under nitrogen limitation
8) Lines 80-82: "In another work, the search for genes.... by transposon mutagenesis." -- How lack of nitrogen benefits fermentation? Please discuss.
RE: We have inserted a paragraph with brief description of the role of yeast assimiliable nitrogen (YAN) in alcohol fermentation in a section above explaining also the need for highly nitrogen efficient strains
9) Line 83: "Deletion of ECM33 gene..." What is the function of ECM33 in yeast. Why is the fermentation time reduced when it is deleted?.. Please clarify
RE: we have inserted a paragraph with more detailed discussion of the possible functions of ECM33 during fermentation.
10) Line 84:.... cell wall damage and osmosis resistance". How cell wall damage and osmosis resistance affects wine-making? Please elaborate.
RE: This phrase is incorrect regarding deltaecm33 strain and was deleted. This strain in fact IS MORE RESISTANT to cell-wall damage and osmotic shock due to increased chitin synthesis. It is only hypersensitive to dyes inhibiting chitin biosynthesis.
11) Line 92: "Deletion of KNR4 gene..." Explain the role of KNR4 in yeast. "to a decrease in protein turbidity under laboratory conditions." Elaborate the conditions.
RE: we have inserted a paragraph explaining the role of KNR4, detailed laboratory conditions used to evaluate the effects of Hpfs overexpression,
12) Please provide a table stating all the yeast strains and their important genotypes and phenotypes for wine-making.
RE: Additional table 2 is provided (see above)
11) Line 101: were introduced into wine strains under control of various promoter.." Please elaborate what promoters were used in previous studies.
RE: In the revised verison promoters used in this study are described.
12) Line 107: "poses" spelling is wrong should be "possess"
RE: changed to “presents”
13) Line 121: "for peptides derived from..." please elaborate on the different peptides that in the past derived from endogenous yeast.
RE: to our knowledge AMP1 and AMP2/3 are the only precisely characterized endogeneous S.cerevisiare peptides with fungi static and fungicidal acitivty against wine-spoilage yeast. We have introduced a section describing these peptides in more detail and also extended the description of killer toxins produced by Sacharomyces and non-saccharomyces yeast species.
14) Line 145: What is crabtree effect? Please discuss
RE: Crabtree effect is briefly explained
15) Lines 158: "glucose transporter HXT2 and regulator MIG1..." proteins need to be written as Hxt2 and Mig1
RE: corrected
16) Line 235: "Rasberry" spelling is wrong should be "Raspberry". The spelling also needs to be corrected in Figure 1. Figure 1 texts needs to be bigger as it is difficult to read.
RE:corrected
17) Line 247: ...promoter of FBA1 gene" -- elaborate the role of FBA1 in yeast.
RE: desciption of FBA1 gene function in yeast is added.
18) Line 250: ....the threshold level of its sensory detection"... Please specify the sensory threshold value in the text
RE: frambion production levels and threshold levels are indicated
19) Line 260: References [59.60] should be [59,60].
RE: corrected

Round 2
Reviewer 3 Report
The authors have answered satisfactorily to all my queries and made changes accordingly.